# Hypoxia-Induced Epithelial-to-Mesenchymal Transition in Proximal Tubular Epithelial Cells through miR-545-3p–TNFSF10

**DOI:** 10.3390/biom11071032

**Published:** 2021-07-15

**Authors:** Mei-Chuan Kuo, Wei-An Chang, Ling-Yu Wu, Yi-Chun Tsai, Ya-Ling Hsu

**Affiliations:** 1Division of Nephrology, Kaohsiung Medical University Hospital, Kaohsiung 807, Taiwan; mechku@kmu.edu.tw; 2Division of Pulmonary and Critical Care Medicine, Kaohsiung Medical University Hospital, Kaohsiung 807, Taiwan; 960215kmuh@gmail.com; 3School of Medicine, College of Medicine, Kaohsiung Medical University, Kaohsiung 807, Taiwan; 4Graduate Institute of Clinical Medicine, College of Medicine, Kaohsiung Medical University, Kaohsiung 807, Taiwan; esther906@gmail.com; 5Division of General Medicine, Kaohsiung Medical University Hospital, Kaohsiung 807, Taiwan; 6Liquid Biopsy and Cohort Research Center, Kaohsiung Medical University, Kaohsiung 807, Taiwan; 7Drug Development and Value Creation Research Center, Kaohsiung Medical University, Kaohsiung 807, Taiwan; 8Graduate Institute of Medicine, College of Medicine, Kaohsiung Medical University, Kaohsiung 807, Taiwan

**Keywords:** hypoxia, miR-545-3p, TNFSF10, epithelial-to-mesenchymal transition, proximal tubular epithelial cells, transcriptome analysis

## Abstract

Hypoxia is regarded as one of the pathophysiologic mechanisms of kidney injury and further progression to kidney failure. Epithelial-to-mesenchymal transition (EMT) in kidney tubules is a critical process of kidney fibrosis. This study utilized transcriptome analysis to investigate hypoxia-induced EMT through microRNA (miRNA)-modulated EMT in proximal tubular epithelial cells (PTECs). RNA sequencing revealed eight miRNAs were upregulated and three miRNAs were downregulated in PTECs cultured under hypoxia compared with normoxia. Among the 11 miRNAs, miR-545-3p has the highest expression in PTECs exposed to hypoxia, and miR-545-3p suppressed tumor necrosis factor-related apoptosis-inducing ligand (TRAIL/TNFSF10) expression. Hypoxia induced EMT in PTECs through miR-545-3p–TNFSF10 modulation, and TNFSF10-attenuated EMT resulted from hypoxia or miR-545-3p mimic transfection. These findings provided new perceptions of the unique regulation of the miR-545-3p–TNFSF10 interaction and their potential therapeutic effect in kidney injury induced by hypoxia.

## 1. Introduction

Kidney disease is a global health issue, which increases morbidity and mortality, and further worsens heavy financial burdens worldwide. Kidney fibrosis has been a landmark of poor progression of various kidney diseases, leading to end-stage kidney disease [1].

Hypoxia plays a critical role in kidney tissue injury and further progression to kidney fibrosis [2]. Hypoxia is a potent regulator of multiple cellular processes, including metabolism, growth, and cell survival via the mechanisms of oxygen sensing [3]. The transcriptional responses to oxygen deprivation are mainly mediated by hypoxia-inducible factors (HIF), which operate during normal development and in pathological processes in association with decreased oxygen availability [4]. Chronic hypoxia can trigger a kidney damage response and cause irreversible phenotype transformation in tubular epithelial cells, inducing renal fibrosis [5,6]. Epithelial-to-mesenchymal transition (EMT) is a critical process of kidney fibrosis by which epithelial cells lose their epithelial characteristics and acquire the mesenchymal features [7,8]. The activation of HIF signaling in renal epithelial cells may promote fibrogenesis by facilitating EMT [9]. Alteration in oxygen levels and hypoxic signaling activation through HIF are emerging as important triggers and modulators of EMT [10]. However, the molecular mechanisms underlying hypoxia inducing EMT and fibrosis in kidney are not fully understood.

MicroRNAs (miRNAs), as post-transcription regulatory factors, are regarded as powerful regulators of gene expression and cellular phenotypes. Accumulating evidence shows hypoxia modulates the biogenesis and activity of miRNAs [11]. Some studies have reported different expressions of miRNAs and their impacts on development of EMT and fibrosis in kidneys under hypoxia [12,13,14]. Du et al. suggested that downregulation of miR-34a promoted EMT in tubular epithelial cells [13]. Xie et al. indicated that upregulation of miR-155 resulted in fibrosis in proximal tubules [14]. However, whether miRNAs mediate the pathogenetic signaling pathway of hypoxia-induced EMT in proximal tubular epithelial cells (PTECs) has not been well-explored by using transcriptome analysis.

Thus, in this study, we used small RNA sequencing analysis of PTECs under normoxia and hypoxia conditions to investigate the potential mechanisms of regulating the hypoxia-mediated signaling pathways of kidney injury.

## 2. Materials and Methods

### 2.1. Cell Culture and Morphology Observation

Human PTECs (ATCC PCS-400-010) were cultured in renal epithelial cell basal medium (ATCC PCS400030™) plus 0.5% fetal bovine serum (FBS), according to the manufacturer’s suggestion. HK-2 cells were purchased from the American Type Culture Collection (Manassas, VA, USA). HK-2 cells were cultured in keratinocyte-SFM (GIBCO) with supplemented 2% FBS. Cells were cultured under normoxia (O_2_, 21%) and hypoxia (O_2_, 1%).

The characteristics of EMT in human PTECs were identified by serial observations of their morphological changes using light microscopy (Nikon ECLIPSE TE20000-S, Nikon, Tokyo, Japan).

### 2.2. Western Blot Analysis

The total protein of HK-2 cells was extracted using RIPA (radio-immunoprecipitation assay) lysis buffer (EMD Millipore, Burlington, MA, USA). The denatured protein was separated by 9–11% SDS-PAGE electrophoresis, and then transferred onto a PVDF membrane following blocking and immunoblotting by specific primary and secondary antibodies. Antibodies against HIF-1α (Catalog #nb100-105, Novus), HIF-2α (Catalog #7096s, Cell Signaling), N-cadherin (Catalog #610921, BD Biosciences), vimentin (Catalog #550513, BD Biosciences), E-cadherin (Catalog #610182, BD Biosciences), slug (Catalog #9585s, Cell Signaling), and GAPDH (Catalog #MAB374, EMD Millipore) were utilized. The signals of the blots were captured using the Proteinsimple+Fluorchem Q system (Alpha Innotech, San Leandro, CA, USA). Densitometry of the blots was calculated using Image J software (Bethesda, MD, USA).

### 2.3. Small RNA Sequencing

HK-2 cells were cultured under hypoxia or normoxia conditions for 48 h. NGS was performed to examine miRNA profiles of HK-2 cells. Trizol^®^ Reagent (Invitrogen, Waltham, MA, USA) was utilized to extract total RNA from the harvested cells, which were isolated for further RNA preparation and small RNA-seq by Welgene Biotechnology Company (Welgene, Taipei, Taiwan). The quality of the extracted RNA was evaluated by an RNA integrity number (RIN), which was measured using an Agilent Bioanalyzer (Agilent Technology, Santa Clara, CA, USA). Samples were prepared to manufacture the small RNA library and then to execute deep sequencing by the Illumina sample preparation kit. PCR amplification was performed to ligate total RNA with 3′ and 5′ adaptors and to reverse-transcription into cDNA. cDNA constructs were separated using 6% polyacrylamide gel electrophoresis, and 18–40 nucleotide RNA fragments (140–155 nucleotide in length with both adapters) were extracted. The sequencing of the libraries was performed using an Illumina GAIIx instrument (50 cycle single read) and then the results were processed by the Illumina software. The differentially expressed miRNAs between HK-2 cells treated with normoxia or hypoxia were defined at >2-fold change and >10 reads per million.

### 2.4. Data Availability

The RNA sequencing data generated in this publication have been deposited in GEO under the accession GSE178810.

### 2.5. RNA Isolation, Reverse Transcription, and Quantitative Real Time PCR (Q-PCR)

Total RNA from cells cultured under normoxia or hypoxia conditions or treated with HIF-1 inhibitor (CAY10585, (3-(2-(4-Adamantan-1-yl-phenoxy)-acetylamino)-4-hydroxybenzoic acid methyl ester) for 48 h (10 uM, catalog #400092, Calbiochem) was isolated using TRIzol and TRIzolLS Reagent (Life Technologies), respectively. miRNAs were reverse-transcribed using the Mir-X™ miRNA First Strand Synthesis Kit (Catalog #638313 Takara, Japan). SYBR Green was used to analyze the quantitative miRNA with a QuantStudio 3Q-PCR system (ThermoFisher Scientific, Foster City, CA, USA). Relative expression levels of the miRNA and RNA in cells were normalized to the internal control U6 or GAPDH, respectively. Relative expressions were presented using the 2^−ΔΔCt^ method. The primers used are listed in Appendix A.

### 2.6. Bioinformatics Analysis Tools

The targets of specific miRNAs were predicted using two bioinformatics websites, the miRmap [15] and TargetScan [16] databases, which can provide miRNA target predictions for different organisms. Both miRmap and TargetScan software classifies potential specific miRNA targets and indicate the repression strength of a miRNA target based on the miRmap scores and percentiles of Context++ score [17].

The biologic functions of the selected miRNA targets were analyzed using IPA software (Ingenuity Systems, Redwood City, CA, USA), which provides a “core analysis” of the genes/proteins. Canonical pathway, network, and toxicity lists obtained from the core analysis can be used to identify the potential pathogenesis of the genes/proteins [18]. 

### 2.7. Transient Transfection

The miR-545-3p mimic (200 nM) and miR-negative control of the mimic (miR-NC, 200 nM) (GE Healthcare, USA) were transfected into cells using the Lipofectamine™ RNAiMAX transfection reagent (Catalog #13778075, ThermoFisher Scientific, USA) following the manufacturer’s protocols. The sequence of the miR-545-3p mimics used are listed in Appendix A.

### 2.8. Enzyme-Linked Immunosorbent Assay (ELISA)

The TNFSF10 concentrations of the supernatants of the HK-2 cells cultured under normoxia, hypoxia, or after transfection with the miR-545-3p mimic under hypoxia conditions for 48 h were measured with the commercially available Quantikine1 ELISA kit (Catalog #DTRL00, R&D Systems) according to the manufacturer’s instructions, as previously described.

### 2.9. Statistical Analysis

The continuous variables were expressed as the mean ± standard error of the mean (SEM) or median (25th and 75th percentile) as appropriate, while categorical variables were expressed as percentages. The correlation among continuous variables was examined by Spearman correlation. The significance of the differences in continuous variables between groups was tested using Student’s *t*-test or one-way analysis of variance (ANOVA), followed by the post hoc test adjusted with a Tukey correction as appropriate. Statistical analyses were conducted using GraphPad Prism 5.0 (GraphPad Software Inc., San Diego, CA, USA). Statistical significance was set at a two-sided *p*-value of <0.05. 

## 3. Results

### 3.1. Hypoxia Induces EMT in PTECs

Hypoxia has been reported to participate in pathophysiologic mechanisms of kidney injury [9]. To investigate the phenotypic alteration of human renal PTECs (RPTECs) and HK-2 cells under hypoxia conditions, cells were cultured under normoxia (O_2_, 21%) and hypoxia (O_2_, 1%) conditions for 48 h. We found that HIF-1α and HIF-2α expression were elevated in RPTECs and HK-2 cells under hypoxia conditions for 6 h (Figure 1A). The cell morphologies were observed and changed from a round shape to an elongated and motile phenotype under hypoxia conditions compared to normoxia conditions (Figure 1B). Western blotting was used to evaluate EMT in PTECs under normoxia or hypoxia conditions. Hypoxia elevated N-cadherin and vimentin expression, and reduced E-cadherin expression in human PTECs (Figure 1C) and HK-2 cells (Figure 1D) after a 48-h incubation period. Therefore, hypoxia causes EMT in PTECs.

### 3.2. Identification of miR-545-3p Participating in Hypoxia-Induced EMT in PTECs

The flow chart of exploration of potential miRNAs induced by hypoxia is shown in Figure 2A. The profile of small RNAs from HK-2 cells cultured under normoxia or hypoxia for 24 h were profiled by NGS. Twenty-one miRNAs with a significant 2-fold change were found in HK-2 cells exposed to hypoxia compared with those cultured under normoxia. Of the 21 miRNAs, 10 miRNAs were upregulated and 11 miRNAs were downregulated. After excluding the miRNAs with a raw read count ≤ 10, 11 significant miRNAs (8 upregulated and 3 downregulated under hypoxia conditions) were displayed by heat mapping (Figure 2B and Appendix A). We used RT-Q-PCR to validate the expression of these miRNAs in two in vitro models, including human RPTECs and HK-2 cells under normoxia and hypoxia conditions. Among the 11 miRNAs, miR-190a-3p and miR-1277-3p could not be detected by qT-PCR, and the inconsistent expression of miR-1269a, miR-3613-3p, miR-33b-5p, miR-33a-5p, and miR-219a-5p were found in human PTECs and HK-2 cells after treatment with hypoxia (Figure 2C–G). The miR-1266-5p, miR-4474-3p, and miR-5579-3p levels were decreased in both human PTECs and HK-2 cells under hypoxia conditions (Figure 2H–J). miR-545-3p expression not only was the highest among the 11 miRNAs in HK-2 cells exposed to hypoxia compared with those treated with normoxia, but also was elevated in human PTECs under hypoxia conditions (Figure 2K). We further examined whether HIF-1 is involved in regulation of miR-545-3p expression (Appendix A), and found that HIF-1 inhibitor suppressed miR-545-3p elevation in HK-2 cells induced by hypoxia (Figure 2L). The above findings indicated that HIF-1 regulated miR-545-3p expression in the proximal tubules under hypoxia conditions.

According to the highest expression of miR-545-3p among the 11 miRNAs in HK-2 cells exposed to hypoxia, the pathophysiologic function of the miR-545-3p-based potential targets, with a miRmap score ≥ 90 according to miRmap database, was analyzed using a core analysis of the IPA database. The top ten of the canonical pathways of the IPA analysis revealed that the pathophysiologic function of miR-545-3p included regulation of EMT by growth factors (Figure 3A). The tox list analysis indicated that miR-545-3p was correlated with renal injury and oxidative stress response (Figure 3B). Based on results of the bioinformatics analysis, hypoxia might induce EMT in PTECs through miR-545-3p modulation.

### 3.3. TNFSF10 as a Direct Target of miR-545-3p

We further investigated the potential downstream target of miR-545-3p, which participates in the hypoxia-mediating EMT in PTECs. Among the predicted targets of miR-545-3p, TNFSF10 participated in EMT regulation (Table 1) and renal injury (Table 2).

In addition, network analysis of the potential targets of miR-545-3p displayed that miR-545-3p and TNFSF10, as targets regulated by miR-545-3p, were also involved in cellular proliferation and the cell cycle (Table 3).

miRmap and TargetScan (version 7.1) were utilized to conduct bioinformatic predictions, which indicated the 3′ UTR of TNFSF10 contained the target seed sequence for the binding of miR-545-3p. The probabilistic miRmap scores indicate the probability of the predicted target genes of miR-545-3p (Figure 3C). The species-conserved miR-545-3p seed sequence in the 3′ UTR of TRAIL is shown in Figure 3D. Hypoxia treatment decreased the TNFSF10 mRNA level of human PTECs (Figure 3E) and HK-2 cells (Figure 3F). Besides, a decreased TNFSF10 mRNA level was found in supernatants derived from HK-2 cells under hypoxia compared to normoxia (Figure 3G). Furthermore, transfection of the miR-545-3p mimic reduced TNFSF10 mRNA expression in the supernatant of HK-2 cells (Figure 3H). HIF-1 inhibitor reversed the decreased TNFSF10 mRNA expression in HK-2 cells induced by hypoxia (Figure 3I). Thus, hypoxia regulated the expression of miR-545, which subsequently decreased the TNFSF10 expression, and HIF-1 might modulate the miR-545-3p–TNFSF10 pathway.

### 3.4. Hypoxia Induces EMT in HK Cells by miR-545-3p–TNFSF10 Modulation

To assess whether miR-545-3p modulated hypoxia-inducing EMT in the proximal tubule, we assessed the biologic effects of miR-545-3p and TNFSF10 in HK-2 cells. Firstly, treatment with TNFSF10 (20 ng/mL) did not affect the cell viability of HK-2 cells (Figure 4A). We further examined slug expression, as one of transcription factors for EMT regulation. Slug expression was increased in HK-2 cells after transfection with miR-545-3p (Figure 4B) and was suppressed in HK-2 cells treated with TNFSF10 (Figure 4C) under normoxia conditions. After transfection of the miR-545-3p mimic, E-cadherin expression was downregulated, and N-cadherin and vimentin expression were upregulated in HK-2 cells under normoxia conditions (Figure 4D). However, TNFSF10 reversed the effect of miR-545-3p in EMT induction in HK-2 cells. In addition, treatment with TNFSF10 (20 ng/mL) prevented E-cadherin downregulation and suppressed N-cadherin and vimentin upregulation, and reversed EMT in HK-2 cells induced by hypoxia in HK-2 cells (Figure 4E). These findings meant that hypoxia contributed to EMT in PTECs through modulation of the miR-545-3p–TNFSF10 pathway, and TNFSF10 could attenuate EMT in PTECs induced by hypoxia and miR-545-3p.

## 4. Discussion

Hypoxia causes morphologic and pathophysiologic dysfunction of the kidney, leading to a rapid decline in kidney function [16]. This study used transcriptome analysis to investigate the potential miRNAs participating in the hypoxia-induced EMT process in PTECs, and demonstrated that miR-545-3p was upregulated in PTECs treated with hypoxia, and HIF-1 modulated miR-545-3p and TNFSF10 expression. miR-545-3p upregulation led to EMT in PTECs under hypoxia. Furthermore, TNFSF10 ameliorated EMT in PTECs treated with hypoxia. Therefore, we have obtained new perceptions by realizing the unique regulation of miR-545-3p–TNFSF10, from which we can interpret kidney disease progression (Figure 5).

Our findings demonstrated that hypoxia regulated miR-545-3p expression in proximal tubules. miRNAs are known to play a critical role in modulation of gene expression or activity and further regulate the pathophysiologic mechanisms of onset or progression of diseases [19]. Previous studies revealed miR-545 served as a tumor promoter, which regulated cell proliferation, invasion, and migration, further resulting in hepatocellular carcinoma [20,21]. miR-545 induced an inflammation process and contributed to hepatitis B-related liver cirrhosis through targeting Tim-3 [22]. Conversely, miR-545-3p partially blocked the lncRNA XIST and enhanced the apoptosis of cardiac myoblasts induced by hypoxia/reoxygenation [23]. However, whether miR-545 participates in the pathogenesis of kidney diseases is not well explored. A few studies demonstrated that miR-545-3p participated in sepsis-related acute kidney injury [24,25]. Tan et al. reported that circ_0091702 serves as a sponge of miR-545-3p to suppress HK-2 cells apoptosis induced by lipopolysaccharide (LPS) through thrombospondin 2 upregulation [24]. Hu et al. found that long non-coding Cancer Susceptibility 2 overexpression relieved LPS-induced apoptosis in HK-2 cells through regulating miR-545-3p/peroxisome proliferator-activated receptor-α axis [25]. Shi et al. indicated that circPRKCI rescued inflammatory injury induced by LPS in HK-2 cells by suppressing miR-545/zinc finger E-box-binding homeobox 2 [26]. This present study firstly indicated the impact of miR-545-3p on hypoxia-induced EMT in the kidney. Hypoxia elevated the miR-545-3p levels, and HIF-1 regulated miR-545-3p expression in PTECs. miR-545-3p induced slug elevation and further promoted EMT, including suppression of miR-545-3p E-cadherin expression and enhancement of N-cadherin and vimentin expression in PTECs. We demonstrated a novel signal pathway of modulating the EMT process in proximal tubules under hypoxia.

Hypoxia also has been reported to modulate TNFSF10 expression and the physiologic process [27,28]. Fang et al. suggested that HIF-1α increased TNFSF10-induced apoptosis via increasing TNFSF10 decoy receptor 1 (DcR1) expression in traumatic brain injury [27]. Harashima et al. indicated that HIF-2α increased susceptibility of pancreas cancer cells to TNFSF10 under hypoxia condition [28]. Nevertheless, it is not clear how to regulate TNFSF10 through post-transcription of miRNAs, especially in hypoxia-induced kidney injury. Our findings provide new insight in that hypoxia modulates TNFSF10 expression in the proximal tubules through miR-545-3p. Hypoxia augmented the miR-545-3p levels, and, in turn, suppressed TNFSF10 expression in PTECs and their supernatants to induce the EMT process. In addition, we also found that hypoxia modulated the miR-545-3p–TNFSF10 pathway in proximal tubules through HIF-1. However, whether HIF-2 regulates this pathway in kidney under a hypoxia environment is not clear. Further study is necessary to explore the impact of the subtypes of the HIF transcription factors on miR-545-3p–TNFSF10 regulation in kidney injury. 

Accumulating evidence shows that the TNF superfamily members are actively implicated in the pathophysiology of the development and progression of kidney diseases [29]. The TNF superfamily members regulate cell performance, such as differentiation, proliferation, necrosis, apoptosis, or fibrosis, dependent on the different stages of kidney damage [30,31]. TNFSF10, as a potential anti-cancer therapeutic agent, could inhibit cell growth and enhances cell apoptosis via binding to specific type I transmembrane death receptors, thereby assisting in the efficacy of the chemotherapy treatment of various cancers [29,32]. Recent studies found the negative relationship between serum TNFSF10 and clinical outcomes in diseases of non-cancers [33,34]. However, the role of TNFSF10 in kidney diseases has not been fully explored. Elevated expression of circulating TRAIL was found in patients with some kidney diseases, such as diabetic kidney disease and minimal change disease [35,36]. Inhibiting TNFSF10 attenuated damage in an in vivo model of ischemia–reperfusion kidney [37]. Conversely, circulating TNFSF10 was decreased in patients with autosomal dominant polycystic kidney disease compared with normal individuals [38]. Otherwise, the inconsistent effects of TNFSF10 on kidney diseases also have been reported. In contrast to the apoptotic effect on PTECs in diabetic nephropathy [30], Nguyen et al. reported that TNFSF10 exerted an inflammatory response and cell proliferation in lupus nephritis [39]. These contradictory statements might be related to different regulation of TNFSF10 among various kinds and stages of kidney diseases. Up to date, it is still difficult to conclude about the actual role of TNFSF10 in the development and progression of kidney diseases. This present study found that TNFSF10 did not affect PTEC viability, suggesting that it did not induce apoptosis in this type of cell. Overexpression of TNFSF10 could ameliorate EMT induced by hypoxia in proximal tubules, suggesting the potential therapeutic effect of miR-545-3p–TNFSF10 in hypoxia-related kidney injury. 

## 5. Conclusions

This study demonstrated that hypoxia contributed to EMT through miR-545-3p–TNFSF10 modulation, and miR-545-3p inhibition and TNFSF10 reversed hypoxia-induced EMT in PTECs. Therefore, these findings provide new insights into the unique regulation of miR-545-3p–TNFSF10 and their potential therapeutic effect in kidney injury induced by hypoxia.

## Figures and Tables

**Figure 1 biomolecules-11-01032-f001:**
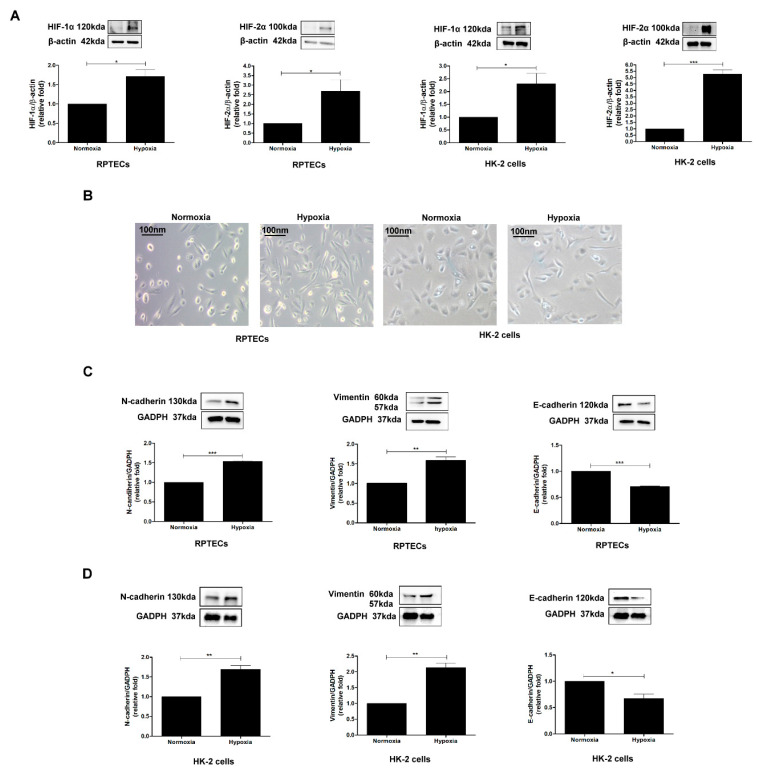
Hypoxia induces EMT in PTECs. (**A**) HIF-1α and HIF-2α expression were assessed in RPTECs and HK-2 cells were incubated under normoxia (O_2_, 21%) and hypoxia (O_2_, 1%) conditions for 6 h using Western blotting. (**B**) The effect of hypoxia on morphological changes in RPTECs and HK-2 cells under normoxia and hypoxia conditions for 48 h. EMT markers (E-cadherin, N-cadherin, and vimentin) were examined in human RPTECs (**C**) and HK-2 cells (**D**) treated with normoxia and hypoxia for 48 h using Western blotting. The bar graph represents the mean ± SEM of at least three independent experiments. * *p* < 0.05, ** *p* < 0.01, *** *p* < 0.001 by Student’s *t*-test.

**Figure 2 biomolecules-11-01032-f002:**
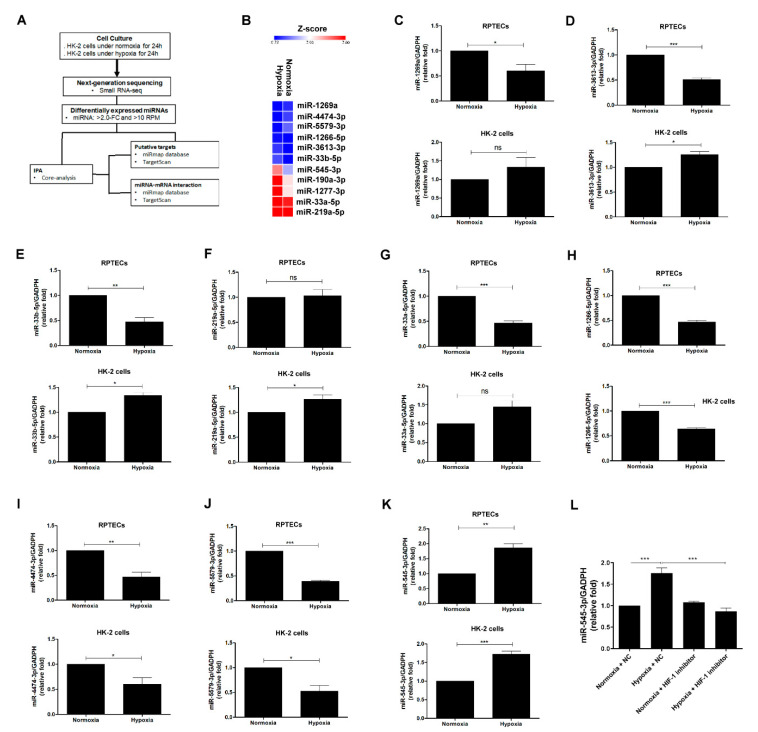
Identification of miR-545-3p participating in hypoxia-induced EMT in PTECs. (**A**) Flowchart of the identification of potential miRNAs associated with hypoxia-induced EMT in HK-2 cells. (**B**) The heat map displayed differential expression from HK-2 cells under normoxia and hypoxia with Z-score values. miR-1269a (**C**), miR-3613-3p (**D**), miR-33b-5p (**E**), miR-219a-5p (**F**), miR-33a-5p (**G**), miR-1266-5p (**H**), miR-4474-3p (**I**), miR-5579-3p (**J**), and miR-545-3p (**K**) levels in human RPTECs and HK-2 cells cultured under normoxia and hypoxia for 48 h were assessed by quantitative real-time PCR. miR-545-3p expression was also examined in HK-2 cells treated with HIF-1 inhibitor (10 uM) under normoxia and hypoxia for 48 h (**L**). The bar graph represents the mean ± SEM of at least three independent experiments. * *p* < 0.05, ** *p* < 0.01, *** *p* < 0.001 by Student’s *t*-test.

**Figure 3 biomolecules-11-01032-f003:**
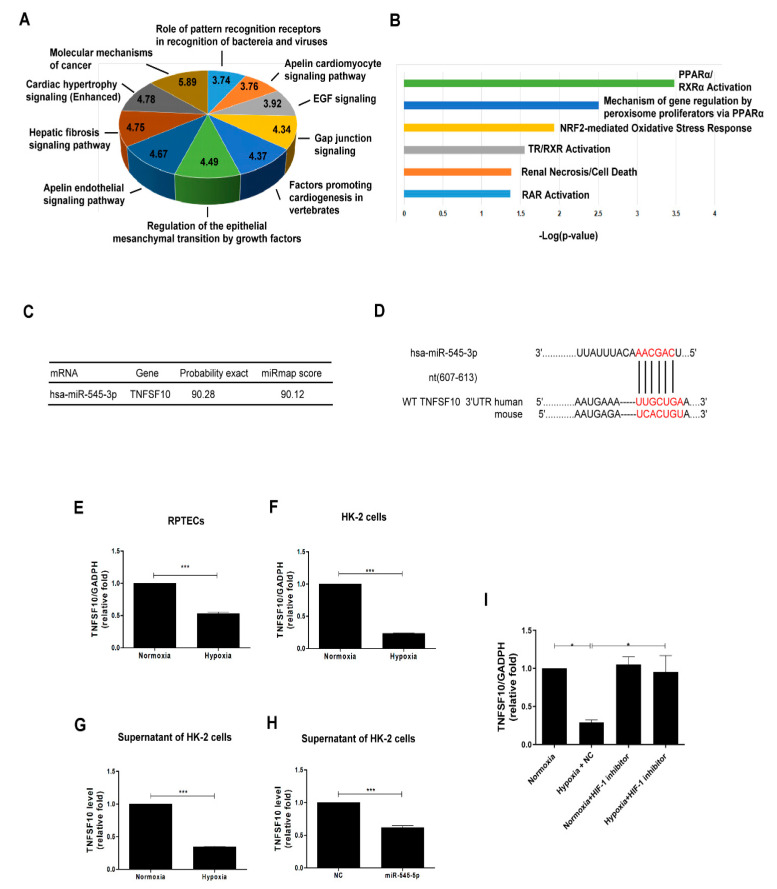
TNFSF10 as a direct target of miR-545-3p. (**A**) Top ten canonical pathways of the potential genes targeted by miR-545-3p using IPA analysis. The pie chart indicates the-Log10 (false discovery rate, FDR) of each canonical pathway. (**B**) Tox list analysis of potential genes targeted by miR-545-3p according to IPA analysis. (**C**) The predictive binding score of miR-545-3p on 3′UTR of TNFSF10 mRNA according to the miRmap database. (**D**) A schematic representation of sequence alignment of TNFSF10 mRNA 3′UTR based on TargetScan version 7.2. TNFSF10 mRNA levels in human RPTECs (**E**) and HK-2 cells (**F**) under normoxia and hypoxia for 48 h were examined using quantitative real-time PCR. The TNFSF10 protein expressions of the supernatants of the HK-2 cells cultured under hypoxia (**G**) or after transfection with miR-545-3p (**H**) for 48 h were assessed by ELISA. TNFSF10 mRNA expression was also examined in HK-2 cells treated with HIF-1 inhibitor (10 uM) under normoxia and hypoxia for 48 h by quantitative real-time PCR (**I**). The bar graph represents the mean ± SEM of at least three independent experiments. * *p* < 0.05, *** *p* < 0.001 by Student’s *t*-test. WT, wild type.

**Figure 4 biomolecules-11-01032-f004:**
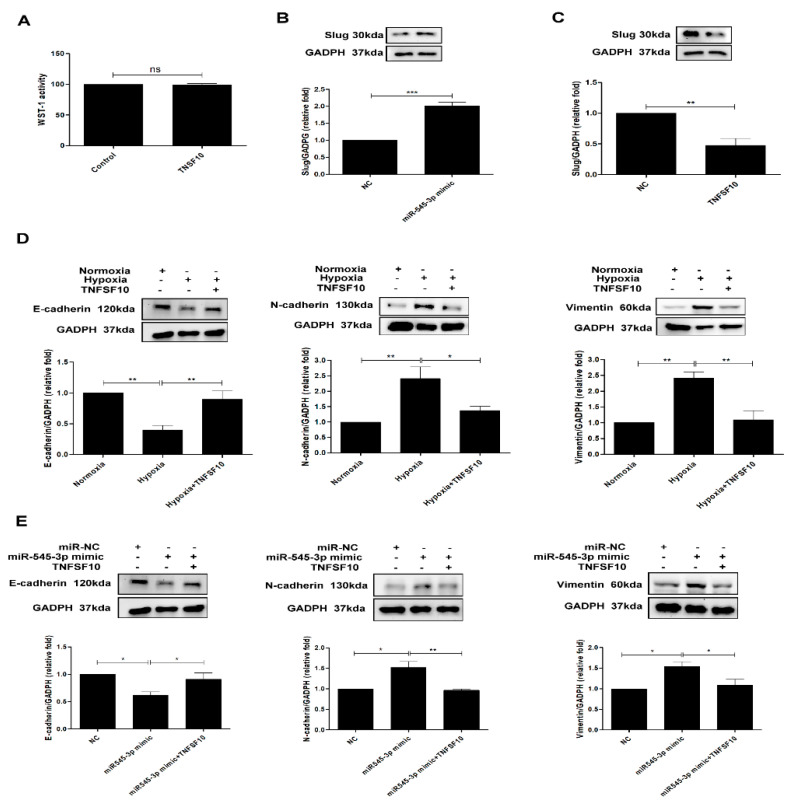
Hypoxia induces EMT in HK-cells by miR-545-3p–TNFSF10 modulation. (**A**) Cell viability activity was examined in HK-2 cells treated with normal control or TNFSF10 (20 ng/mL) for 48 h using the WST-1 assay. (**B**) Slug protein was assessed in HK-2 cells after transfection with the miR-NC (200 nM) or miR-545-3p mimic (200 nM) for 24 h and then cultured under normoxia conditions for 48 h. (**C**) Slug protein was also examined in HK-2 cells treatment with TNFSF10 (20 ng/mL) under normoxia for 48 h. (**D**) EMT markers were examined in HK-2 cells treated with normoxia, hypoxia, and TNFSF10 (20 ng/mL) plus hypoxia for 48 h. (**E**) HK-2 cells were transfected with the miR-NC (200 nM) or miR-545-3p mimic (200 nM), and 24 h after transfection, cells were treated with TNFSF10 (20 ng/mL) for 48 h. EMT was assessed by Western blotting for E-cadherin, N-cadherin, and vimentin. The bar graph represents the mean ± SEM of at least three independent experiments. * *p* < 0.05, ** *p* < 0.01, *** *p* < 0.001 by Student’s *t*-test.

**Figure 5 biomolecules-11-01032-f005:**
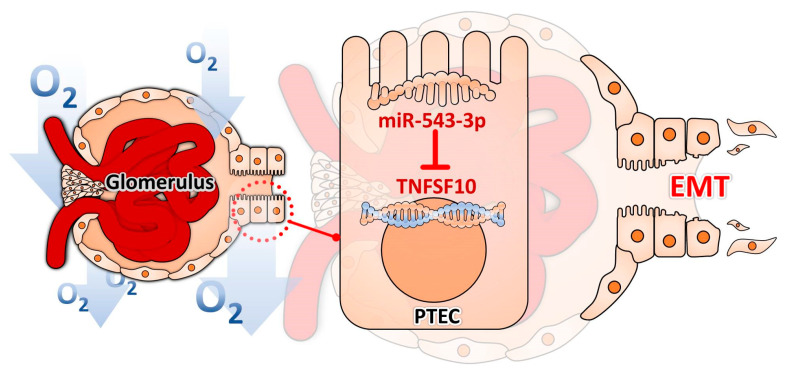
Illustration of the mechanism by which hypoxia induced EMT in PTECs through miR-545-3p–TNFSF10.

**Table 1 biomolecules-11-01032-t001:** Canonical pathway of the predicted target genes of miR-545-3p according to the IPA database.

Ingenuity Canonical Pathways	−Log (*p*-Value)	Ratio	Molecules
Role of Pattern Recognition Receptors in Recognition of Bacteria and Viruses	3.74	0.0764	DDX58, EDA, MAPK8, PIK3C2A, PIK3C3, PRKCA, PRKD3, RNASEL, SYK, TLR8, TNFSF10
Apelin Cardiomyocyte Signaling Pathway	3.76	0.0918	ARNT, GNAI3, MAPK8, MYLK, PIK3C2A, PIK3C3, PRKCA, PRKD3, SLC9A8
EGF Signaling	3.92	0.127	JAK1, MAPK8, PIK3C2A, PIK3C3, PRKCA, RPS6KB1, SOS2
Gap Junction Signaling	4.34	0.0722	CSNK1G3, GNAI3, GRIA4, GRIK3, HTR2A, PIK3C2A, PIK3C3, PPP3R2, PRKAR1A, PRKCA, PRKD3, SOS2, TUBB, TUBB1
Factors Promoting Cardiogenesis in Vertebrates	4.37	0.0828	BMP8B, FZD6, LRP1, LRP6, MAP3K7, MAPK8, MEF2C, MYOCD, PRKCA, PRKD3, SMAD2, WNT7A
Regulation of the Epithelial Mesenchymal Transition by Growth Factors Pathway	4.49	0.0745	FGF2, FGFR1, HGF, JAK1, LATS1, MAP3K7, MAPK8, PDGFB, PIK3C2A, PIK3C3, SMAD2, SMURF1, SOS2, TNFSF10
Apelin Endothelial Signaling Pathway	4.67	0.0965	ARNT, GNA13, GNAI3, GNAO1, MAPK8, MEF2C, PIK3C2A, PIK3C3, PRKCA, PRKD3, RPS6KB1
Hepatic Fibrosis Signaling Pathway	4.75	0.0582	FGF2, FGFR1, FZD6, GNAI3, ITGA2, JAK1, LRP6, MAP3K7, MAPK8, MYLK, PDGFB, PIK3C2A, PIK3C3, PRKAR1A, PRKCA, PRKD3, PTCH1, RPS6KB1, SMAD2, SOS2, WNT7A
Cardiac Hypertrophy Signaling (Enhanced)	4.78	0.0525	EDA, FGF2, FGFR1, FZD6, GNA13, GNAI3, H2BW2, IFNAR1, IL10RA, IL4R, ITGA2, MAP3K7, MAPK8, MEF2C, MYOCD, PDE3B, PIK3C2A, PIK3C3, PPP3R2, PRKAR1A, PRKCA, PRKD3, RPS6KB1, TNFSF10, WNT7A
Molecular Mechanisms of Cancer	5.89	0.0625	BMP8B, E2F7, FZD6, GNA13, GNAI3, GNAO1, H2BW2, ITGA2, JAK1, LRP1, LRP6, MAP3K7, MAPK8, PIK3C2A, PIK3C3, PRKAR1A, PRKCA, PRKD3, PSEN1, PTCH1, SMAD2, SOS2, TFDP1, WNT7A

**Table 2 biomolecules-11-01032-t002:** Tox list of the predicted target genes of miR-545-3p according to the IPA database.

Ingenuity Toxicity Lists	−Log (*p*-Value)	Ratio	Molecules
RAR Activation	1.37	0.0417	DUSP1, GTF2H5, MAPK8, PRKAR1A, PRKCA, PRKD3, SMAD2, TRIM24
Renal Necrosis/Cell Death	1.38	0.0318	CARD8, DUSP1, FGF2, GLS, GNA13, HGF, HSPA1A/HSPA1B, IPPK, LRP6, MAPK8, PRKCA, PSEN1, PTCH1, SLK, TNFSF10, UNC5C, USP14, VPS13A
TR/RXR Activation	1.55	0.0595	PDE3B, PIK3C2A, PIK3C3, STRBP, SYT2
NRF2-mediated Oxidative Stress Response	1.93	0.0472	CUL3, CYP2C19, MAP3K7, MAPK8, PIK3C2A, PIK3C3, PRKCA, PRKD3, UBE2K, USP14
Mechanism of Gene Regulation by Peroxisome Proliferators via PPARα	2.51	0.0745	DUSP1, HSPA1A/HSPA1B, MAP3K7, PDGFB, PRKAR1A, PRKCA, SOS2
PPARα/RXRα Activation	3.48	0.0667	CAND1, CKAP5, CLOCK, CYP2C19, GPD2, MAP3K7, MAPK8, MEF2C, PRKAR1A, PRKCA, SMAD2, SOS2

**Table 3 biomolecules-11-01032-t003:** Network analysis of the predicted target genes of miR-545-3p according to the IPA database.

Top Diseases and Functions	Score	Focus Molecules	Molecules in Network
Cell Cycle, Cell Death and Survival, Free Radical Scavenging	26	21	Ap1, CARD8, CD3, DDX58, ELP1, FGF2, HMGB1, HTR7, IFNBeta, IFN type 1, IKK (complex), IL1, IL10RA, IL12 (complex), IL4R, Immunoglobulin, Jnk, LDL, MAP1LC3, MAP3K7, MAPK8, MYLK, NFkB (complex), PI3K (complex), PP2A, PRKCA, RNASEL, RPS6KB1, RTF1, SETBP1, TFAM, TLR8, TNFSF10, UHMK1, ZC3HAV1

## Data Availability

The RNA sequencing data generated in this publication have been deposited in GEO under accession GSE178810.

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
