# Peer review of "Hypoxia-Induced Epithelial-to-Mesenchymal Transition in Proximal Tubular Epithelial Cells through miR-545-3p–TNFSF10"

_biomolecules, 2021, doi:10.3390/biom11071032_

Round 1
Reviewer 1 Report
It is an interesting to know the new contributors for Hypoxia-induced EMT, but also very important to know how the new contributors affect the already known mechanisms. In these aspects, some concerns are listed below which might be necessary to address in the revision:
- Among the HIF-1 signaling to cause EMT, how is this miRNA’s effect associated ? Does this miR-545-3p-TNFSF10 axis work independent of HIF-1? Or in the HIF-1 inactivated cells, does this axis work to cause EMT?
- For the mechanisms to cause EMT by this miR-545-3p-TNFSF10 axis, which EMT drivers are associated? This mechanistic part would be very important as a manuscript.
- What is the upstream signals or stimulation for miR-545-3p in the chronic kidney disease context? Also does this miRNA have some known functions in the context?
Author Response
Reviewer 1
Comments and Suggestions for Authors
It is an interesting to know the new contributors for Hypoxia-induced EMT, but also very important to know how the new contributors affect the already known mechanisms. In these aspects, some concerns are listed below which might be necessary to address in the revision:
- Among the HIF-1 signaling to cause EMT, how is this miRNA’s effect associated ? Does this miR-545-3p-TNFSF10 axis work independent of HIF-1? Or in the HIF-1 inactivated cells, does this axis work to cause EMT?
ANS: Thank you for your valuable comment. We have measured miR-545-3p expression in HK-2 cells treated with HIF-1 inhibitor under normoxia and hypoxia conditions, and found that HIF-1 inhibitor suppressed hypoxia-induced miR-545-3p elevation. Thus, HIF-1 participated in miR-545-3p-TNFSF10 axis in proximal tubular injury caused by hypoxia. We have added this result and description in the result (Figure 2L) and discuss section (Line 489-493).
- For the mechanisms to cause EMT by this miR-545-3p-TNFSF10 axis, which EMT drivers are associated? This mechanistic part would be very important as a manuscript.
ANS: Thank you for your valuable comment. We have examined slug expression, as one of transcription factors for EMT regulation, in HK-2 cells after transfection with miR-545-3p mimic and treated with TNFSF10 (20ng/ml) respectively. miR-545-3p mimic increased slug expression and conversely, TNFSF10 reduced slug expression. Thus, miR-545-3p-TNFSF10 axis promoted EMT through slug. We have added this result and description in the result (Figure 4B and C) and discuss section (Line 489-493).
- What is the upstream signals or stimulation for miR-545-3p in the chronic kidney disease context? Also does this miRNA have some known functions in the context?
ANS: Thank you for your valuable comment. To our best knowledge, there was no report to examine the role of miR-545-3p in chronic kidney disease. Few studies demonstrated that miR-545-3p participated in sepsis-related acute kidney injury (J Mol Histol. 2021; J Surg Res. 2021). Tan et al. reported that circ_0091702 serves as a sponge of miR-545-3p to suppress HK-2 cells apoptosis induced by LPS through THBS2 upregulation. Hu et al. found that long non-coding CASC2 overexpression relieved LPS-induced apoptosis in HK-2 cells through regulating miR-545-3p/PPARA axis (J Surg Res. 2021). Previous studies revealed miR-545 served as a tumor promoter, which regulated cell proliferation, invasion and migration, further resulting hepatocellular carcinoma (Biomed Pharmacother 2018, 108, 347-354; Oncotarget 2016, 7, 25350-65). In addition, miR-545 induced inflammation process and contributed to hepatitis B-related liver cirrhosis through targeting Tim-3 (J Cell Biochem 2018). We have revised 2nd paragraph of discussion section (Line 471-488).

Reviewer 2 Report
I read the manuscript entitled "Hypoxia-induced epithelial-to-mesenchymal transition in proximal tubular epithelial cells through miR-545-3p-TNFSF10" submitted to the biomolecules for possible publication.
The authors argued that they demonstrated that hypoxia contributed to EMT through miR-545-3p-TNFSF10 modulation. miR-545-3p inhibition and TNFSF10 reversed hypoxia-induced EMT in PTECs in this study.
The hypothesis of this paper is simple, the claims are straightforward, and the data seem to support the hypothesis well, but there are things the authors could do to improve the manuscript.
#1 Involvement of HIFs
Figure 1 shows the data of HIF-1a and HIF-2a protein expression. The meaning of this data is unclear. In particular, experimental data should be presented to clearly discuss the involvement of HIFs in the regulation of miR-545-3p expression.
Such experiments include inhibition of HIF-a expression using siRNA and HIF activation using HIF-a hydroxylase inhibitors.
#2 Data deposit
RNA-seq analysis has been done in this study. The results, including fastq files, need to be registered in a database. Please explicitly indicate which database it was registered in and the accession number.
#3 Ischemia and hypoxia
The authors erroneously treat the terms ischemia and hypoxia as interchangeable concepts. Hypoxia and ischemia are different concepts. If the authors want to discuss ischemia, they should provide experimental data under low oxygen + low glucose + low aminoacid conditions.
#4 RNA isolation
The authors clearly describe that "Total RNA from cells and extracellular vesicles in the urine was isolated using TRIzol and TRIzolLS Reagent (Life Technologies) respectively" in section 2.4 RNA isolation, reverse transcription, and quantitative real-time PCR (Q-PCR).
However, I cannot find the experimental data using RNA from extracellular vesicles in the urine.
Author Response
Reviewer 2
Comments and Suggestions for Authors
I read the manuscript entitled "Hypoxia-induced epithelial-to-mesenchymal transition in proximal tubular epithelial cells through miR-545-3p-TNFSF10" submitted to the biomolecules for possible publication.
The authors argued that they demonstrated that hypoxia contributed to EMT through miR-545-3p-TNFSF10 modulation. miR-545-3p inhibition and TNFSF10 reversed hypoxia-induced EMT in PTECs in this study.
The hypothesis of this paper is simple, the claims are straightforward, and the data seem to support the hypothesis well, but there are things the authors could do to improve the manuscript.
#1 Involvement of HIFs
Figure 1 shows the data of HIF-1a and HIF-2a protein expression. The meaning of this data is unclear. In particular, experimental data should be presented to clearly discuss the involvement of HIFs in the regulation of miR-545-3p expression.
Such experiments include inhibition of HIF-a expression using siRNA and HIF activation using HIF-a hydroxylase inhibitors.
ANS: Thank you for your valuable comment. We have measured miR-545-3p expression in HK-2 cells treated with HIF-1 inhibitor under normoxia and hypoxia conditions, and found that HIF-1 inhibitor suppressed hypoxia-induced miR-545-3p elevation. Thus, HIF-1 participated in miR-545-3p-TNFSF10 axis in proximal tubular injury caused by hypoxia. We have added this result and description in the result (Figure 2L) and discuss section (Line 489-493).
#2 Data deposit
RNA-seq analysis has been done in this study. The results, including fastq files, need to be registered in a database. Please explicitly indicate which database it was registered in and the accession number.
ANS: Thank you for your suggestion. We have upload RNA-seq analysis to GEO database (GSE178810).
#3 Ischemia and hypoxia
The authors erroneously treat the terms ischemia and hypoxia as interchangeable concepts. Hypoxia and ischemia are different concepts. If the authors want to discuss ischemia, they should provide experimental data under low oxygen + low glucose + low amino acid conditions.
ANS: Thank you for your correction. The aim of this study is to examine the impact of hypoxia on EMT in proximal tubules. We have deleted ischemia to avoid readers misunderstanding.
#4 RNA isolation
The authors clearly describe that "Total RNA from cells and extracellular vesicles in the urine was isolated using TRIzol and TRIzolLS Reagent (Life Technologies) respectively" in section 2.4 RNA isolation, reverse transcription, and quantitative real-time PCR (Q-PCR).
However, I cannot find the experimental data using RNA from extracellular vesicles in the urine.
ANS: Thank you for your correction. We are very sorry about this error, and have deleted extracellular vesicles in the urine in this paragraph.

Reviewer 3 Report
A nice study with novel results.
Some comments can be made:
- at the end of the introduction the main results are presented. That can be deleted.
- line 149 contains false words
- I think a novel study on mir545-3p should be cited:
- as TNFSF10 mainly induces apoptosis: Was apoptosis analyzed in the study? I think that an additional apoptosis assay has to be performed to interpret the results correctely or the term of apoptosis has to be discussed in mre detail
Author Response
Reviewer 3
A nice study with novel results.
Some comments can be made:
- at the end of the introduction the main results are presented. That can be deleted.
ANS: Thank you for your suggestion. We have deleted the main results in the introduction.
- line 149 contains false words
ANS: Thank you for your correction. We have corrected.
- I think a novel study on mir545-3p should be cited:
lncRNA XIST Knockdown Suppresses Hypoxia/Reoxygenation (H/R)-Induced Apoptosis of H9C2 Cells by Regulating miR-545-3p/G3BP2
ANS: Thank you for your suggestion. We have cited this reference (ref 21).
- as TNFSF10 mainly induces apoptosis: Was apoptosis analyzed in the study? I think that an additional apoptosis assay has to be performed to interpret the results
correctely or the term of apoptosis has to be discussed in mre detail
ANS: Thank you for your valuable comment. We have examined viability of HK-2 cell after treatment with TNFSF10 for 48 h using WST-1 assay and the data shows TNSF10 did not affect the cell viability of HK-2 cells, suggesting it did not induce apoptosis in this type of cells (Figure 4A). Thus, the effect of TNFSF10 might be different among various cell types. We have added this discussion in the discussion section (Line 503-524).

Round 2
Reviewer 2 Report
I have reviewed the responses to my comment (#1).
The authors claimed that they used "HIF-1 inhibitor". but they should have indicated the specific drug name.
Experiments are also needed to clarify which transcription factor, HIF-1 or HIF-2, is specifically involved in this phenomenon.
Experiments using siRNA are one way to achieve this goal.
—
#1 Involvement of HIFs
Figure 1 shows the data of HIF-1a and HIF-2a protein expression. The meaning of this data is unclear. In particular, experimental data should be presented to clearly discuss the involvement of HIFs in the regulation of miR-545-3p expression.
Such experiments include inhibition of HIF-a expression using siRNA and HIF activation using HIF-a hydroxylase inhibitors.
ANS: Thank you for your valuable comment. We have measured miR-545-3p expression in HK-2 cells treated with HIF-1 inhibitor under normoxia and hypoxia conditions, and found that HIF-1 inhibitor suppressed hypoxia-induced miR-545-3p elevation. Thus, HIF-1 participated in miR-545-3p-TNFSF10 axis in proximal tubular injury caused by hypoxia. We have added this result and description in the result (Figure 2L) and discuss section (Line 489-493).
—
Author Response
Reviewer 2
I have reviewed the responses to my comment (#1).
The authors claimed that they used "HIF-1 inhibitor". but they should have indicated the specific drug name.
ANS: Thank you for your suggestion. The drug name of HIF-1 inhibitor was CAY10585 (3-(2-(4-Adamantan-1-yl-phenoxy)-acetylamino)-4-hydroxybenzoic acid methyl ester). We have added in the method section (Line 110-111).
Experiments are also needed to clarify which transcription factor, HIF-1 or HIF-2, is specifically involved in this phenomenon.
Experiments using siRNA are one way to achieve this goal.
—
#1 Involvement of HIFs
Figure 1 shows the data of HIF-1a and HIF-2a protein expression. The meaning of this data is unclear. In particular, experimental data should be presented to clearly discuss the involvement of HIFs in the regulation of miR-545-3p expression.
Such experiments include inhibition of HIF-a expression using siRNA and HIF activation using HIF-a hydroxylase inhibitors.
ANS: Thank you for your valuable comment. We found that HIF-1 inhibitor could suppress elevated expression of miR-545-3p and reversed decreased expression of TNFSF10 induced by hypoxia in HK-2 cells (Figure 2L and 3I), meaning that HIF-1 might modulate miR-545-3p-TNFSF10 pathway in proximal tubule under hypoxia. Because of limited time for response, we apologized that we did not examine whether HIF-2 modulated miR-545-3p-TNFSF10 pathway under hypoxia condition. We have added discussion and limitation in the discussion section (Line 514-518).

Reviewer 3 Report
The paper was signifcantly improved by the aditional comments. I suggest to accept it in present form.
Author Response
Thank you for your valuable comment.